# Obstetric Complications in Women from Sub-Saharan Africa—A Cross-Sectional Study

**DOI:** 10.3390/ijerph191610101

**Published:** 2022-08-16

**Authors:** Laura Gombau-Giménez, Pilar Almansa-Martínez, María Suarez-Cortés, Alonso Molina-Rodríguez, César Leal-Costa, Ismael Jiménez-Ruiz

**Affiliations:** 1Faculty of Nursing, University of Murcia, El Palmar, 30120 Murcia, Spain; 2ENFERAVANZA, Murcia Institute for BioHealth Research (IMIB-Arrixaca), Faculty of Nursing, University of Murcia, El Palmar, 30120 Murcia, Spain

**Keywords:** female genital mutilation, female circumcision, pregnant women, obstetric complications, obstetric outcomes, health consequences

## Abstract

Aim: The aim of this study was to identify and compare birthing complications in women originating from countries where they are at risk (may become victims) of FGM with those of Spanish women, all having given birth from 2012 to 2015 at the “Virgen de la Arrixaca” University Clinical Hospital in Murcia, Spain. Methods: A transversal, observational, quantitative study was carried out, retrospectively, comparing 245 sub-Saharan women originating from countries where FGM is practiced with 490 Spanish women, in terms of obstetric complications. Data collection was performed via electronic clinical records. Results: The sub-Saharan women presented higher rates of intrapartum and emergency caesareans, intense postpartum haemorrhages, concurrent episiotomies and tears (2nd and 3rd degree), failed inductions, and non-progressive labours, and a more severe risk of foetal distress when compared with Spanish women. Conclusions: The fact that the sub-Saharan women originating from countries where FGM is practiced presented a greater number of birthing complications than the Spanish women proves the need for Spanish healthcare professionals to receive training towards cultural competency acquisition, in order to provide a multidisciplinary approach, with standardized action protocols focused fundamentally on prevention.

## 1. Introduction

Annually, more than half a million women die around the world as a result of sexual and reproductive health issues, the latter being the primary cause of death of women of a fertile age [1].

The majority of the approximately 140 million annual births worldwide occur in women who present no risk factors for birthing complications in either the mother or the child [2]. However, each year more than 295,000 women lose their lives around the world due to complications during pregnancy or childbirth. The vast majority of these deaths occur in low-resource settings, and most could be prevented [3]. Sub-Saharan Africa and Southern Asia account for approximately 86% of the estimated global maternal deaths. Sub-Saharan Africa alone accounts for roughly 66% of maternal deaths [3]. Taking into account that in most sub-Saharan countries there are differing degrees to which female genital mutilation (FGM), the greatest cause among traditional practices of harmful consequences for the sexual and reproductive health of women, is prevalent, it could thus be considered one of the factors responsible for such drastic statistical figures. The evidence indicates that there is a higher risk of adverse obstetric outcomes in women who have been subject to FGM compared with those who have not [4,5].

According to the World Health Organization [6], FGM “*is a traditional harmful practice that involves the partial or total removal of external female genitalia or other injury to female genital organs for non-medical reasons*.” Such procedures provide no health benefits whatsoever. The elimination of, or harm to healthy genital organs, can produce severe consequences for the health of those women and girls affected, for both the short and long term, including death. The current estimation of women and girls that have been subjected to some type of FGM worldwide is roughly 200 million. This harmful traditional practice is performed mainly in 30 countries, of which 27 belong to sub-Saharan Africa [7].

In general, migratory movements occur because of unmet basic human needs, natural disasters, or the presence of elements that violate human rights [8]. In this sense, female genital mutilation is one of the most serious human rights violations. The combination of these factors is what drives young sub-Saharan Africans to leave their countries of origin. This population is the main source of current and future migration, according to reports of those intending to initiate migration processes [9]. The settlement in western countries and high birth rates have facilitated the transmission of this traditional practice to the countries that receive migrant populations [10]. According to data from the European Parliament [11], in Europe there are about 500,000 women and girls who have been subjected to female genital mutilation and 180,000 are at risk of becoming victims each year, although these figures are underestimated [10].

Thereby, the aims of the present study were to identify and compare the birthing complications in sub-Saharan women from countries where the risk of FGM exists with those of Spanish women, all of whom had given birth from 2012 to 2015 at the “Virgen de la Arrixaca” University Clinical Hospital (HCUVA) in Murcia, Spain.

## 2. Materials and Methods

### 2.1. Design

A transversal, observational, quantitative study was carried out, retrospectively, and involved an initial descriptive stage and subsequent data analysis. The prevalence of a range of variables in those women exposed to the risk factor, in this case originating from and/or being a national of a country in which FGM is practiced, were compared with those of women of Spanish origin and nationality and as such not exposed to said risk factor.

### 2.2. Study Sample

This comprised of women having given birth at the HCUVA in Murcia (Spain) from 2012 to 2015, inclusive. In order to determine each sample group, firstly, all those women originally from countries at risk of FGM and giving birth in the aforementioned timeframe at the HCUVA (n = 245) were selected as the initial case group. Subsequently, all those women of Spanish origin and nationality who gave birth during the same timeframe at the HCUVA (n = 18.225) were selected as the initial control group. Finally, in order to avoid selection bias, the definitive study groups were determined via propensity score matching, based on the variable “origin”, rendering them comparable and homogenous in terms of those characteristics that might influence the phenomenon being studied (“age” and “number of pregnancies”). Group 1—Cases: women from countries practicing FGM (UNICEF, 2016) who gave birth within the established timeframe (n = 245). Group 2—Control: women of Spanish origin and nationality having given birth in the same time period subject to a 1:2 nearest-neighbour pairing technique based on the co-variables age and number of pregnancies (n = 490). The combined definitive study sample was thus comprised of a total of 735 women.

### 2.3. Data collection, Instruments and Analysis

A list of the patient record numbers was requested from the Codification Department at the HCUVA in Murcia, of all those women who were admitted to the Obstetrics and Gynaecology Service for giving birth from January 2012 to December 2015, inclusive. Subsequently, each patient’s digitalised notes were revised via the SELENE software program. An initial review was made of each patient’s medical admission report and the gynaecological discharge report or midwifery report corresponding to the moment of childbirth, according to the type of birth. Following this, three (3) documents were identified as containing all the data required in order to carry out the intended analysis. These were: “Hoja de Paritorios”, “HCE_PUÉRPERAS” and “Informe de Cuidados de Enfermería al Alta”; in English: “Delivery Room Form”, “EPN_POSTPARTUMS” (EPN: Electronic Patient Notes), and “Nursing Care Discharge Report”.

By way of a Microsoft Excel data table, the study target variables were compiled and organised. Finally, a statistical analysis was performed on the obtained data using the free software R (Vienna, Austria), versions 3.3.1 (©2016), 3.3.3 (R Core Team 2017), 3.4.1 (R Core Team 2017), and Knitr (R Package Version 1), a general use software package for dynamic report generation with R.

### 2.4. Ethical Considerations

In order to gain access to the patient records of the study subjects, permission was requested via a letter to the General Director of Healthcare Area 1 (Área I de Salud) at the HCUVA. The entire research team is committed to the confidentiality of the data obtained and to their use for research purposes alone, in accordance with the Spanish Data Protection Act “Ley Orgánica de Protección de Datos de carácter personal 15/1999 del 13 de diciembre”.

## 3. Results

In the time period studied, no record of FGM was detected in the computerized medical records. Nor were consistent records of genitalia status detected for any of the women included in the study. Despite these results, it was decided to compare the obstetric results between the two population groups to determine whether or not there were significant differences in this variable.

### 3.1. Sociodemographic Characteristics

The present study involved 245 sub-Saharan and 490 Spanish women, and thus a total sum of 735 women. The 245 sub-Saharan women originated from 15 of the 27 countries where FGM is considered a risk according to UNICEF (2016), mainly Nigeria, Senegal, and Mali (Figure 1).

The age of both the sub-Saharan and the Spanish women ranged from 18 to 44 years old. The mean age of the total study sample (N = 735) was 31.51 ± 5.28 years old.

### 3.2. Type of Birth

The predominant type of birth among the entire study sample (N = 735) was eutocic (65.31%), followed by caesarean (20.82%), instrumental delivery (12.38%), and abortion (1.36%). The type of birth was not registered in 0.14% of the total deliveries.

On applying Pearson’s χ^2^ in order to determine whether there is an association between the variables origin and type of birth, the resultant *p*-value was <0.05 (*p* = 1.616 × 10^−9^). Therefore, in general, an association does exist between the participating women’s origin and the type of birth.

Comparing sub-Saharan women with Spanish women in terms of the type of birth (95% CI), abortion did not present with any statistically significant difference (χ^2^ = 2.1416; *p*-value = 0.1433). Regarding eutocic birth, statistically significant differences were found between both groups (χ^2^ = 17.571; *p*-value = 2.768 × 10^−5^), whereby these were notably greater in proportion for Spanish women (70.60%) than for sub-Saharan women (54.69%). Instrumental birth (forceps and vacuum) presented with no statistically significant differences between groups (χ^2^ = 0.18969; *p*-value = 0.6632). Caesarean births were seen in 15.71% of the Spanish women and almost double that, 31.03%, for sub-Saharan women. In regard to the type of caesarean, there were no statistically significant differences between groups for elective caesareans (χ^2^ = 0.10059; *p*-value = 0.7511), although this was the predominant type of caesarean for the Spanish women. Intrapartum caesareans did present with statistically significant differences between groups (χ^2^ = 10.211; *p*-value = 0.001396), with this type of delivery being predominant among the sub-Saharan women. Finally, urgent caesareans also presented with statistically significant differences between the two groups (χ^2^ = 28.254; *p*-value = 1.064 × 10^−7^), with a prevalence of 6.94% in the sub-Saharan women as opposed to 0.20% in the group of Spanish women (Table 1). 

### 3.3. Reasons for Caesareans

Among those women who were subject to caesarean, the reasons for this approach according to origin are shown in Table 2. Foetal breech presentation was the most frequent reason for elective caesarean in the Spanish women. On the other hand, for sub-Saharan women, the main reason was previous caesarean. In regard to the reasons for intrapartum caesarean, failed induction and non-progressive labour were present in greater proportions among sub-Saharan women than among Spanish women. In the case of urgent caesarean, of the 18 cases reported in the present study, only one corresponded to the Spanish group, while the remaining 17 were sub-Saharan, for whom the predominant reasons reported were severe risk of foetal distress (SRFD) and intense haemorrhage.

### 3.4. Reasons for Inducing Labor

The induction of labour was observed to have occurred in 21.90% of the total number of study subjects (N = 735), and intragroup percentages for both samples are very similar. The predominant reason for inducing labour in both groups was premature membrane rupture, although among the Spanish women, the percentage was double that of the sub-Saharan women. In the latter group an almost equally prevalent reason for induction was non-progressive labour (Table 3).

### 3.5. Reasons for Instrumental Delivery

Instrumental births (forceps and vacuum) were observed in 12.38% of the total study sample (N = 735). There were no statistically significant differences between the two groups (χ^2^ = 0.18969; *p*-value = 0.6632), although the predominant reasons for instrumental births in the Spanish women were non-progressive labour and failed induction due to premature membrane rupture, while for the sub-Saharan women the predominant reasons were non-progressive labour and risk of foetal distress (Table 4).

### 3.6. Episiotomy/Tear/Intact Perineum

Regarding the variable episiotomy/tear/intact perineum of the entire study sample (N = 735), 55.93% of the women participants presented with an intact perineum, 19.05% some form of tear, in 20.95% an episiotomy was performed, and in 4.08% both an episiotomy and a tear were observed to have taken place. If the 140 tears (19.05%) alone are added to the 30 (4.08%) tears concurring with an episiotomy, the total number of tears is 170, which is proportionally 23.13% of the total sample. Likewise, if the 154 episiotomies (20.95%), are added to the 30 (4.08%) episiotomies concurring with tears, the total number of episiotomies is 184, which is 25.03% of the total sample. On analysing the data according to origin, it can be seen that in the sub-Saharan women, the proportion of tears and episiotomies plus tears (concurrently in the one same delivery) was substantially greater than for the Spanish women (Table 5). On the other hand, having an intact perineum was proportionally greater for the Spanish women than for the sub-Saharans. Likewise, the percentage of episiotomies was substantially greater for the group of Spanish women. On analysing the variable tear according to type and origin, it is noteworthy that among the total study sample, only seven suffered a third degree tear, all of whom were sub-Saharan (Table 5).

## 4. Discussion

The results of the present study showed obstetric–gynaecological complications during birth were present to a greater degree for sub-Saharan women originally from countries practicing FGM compared with Spanish women. However, it is not certain that the differences are due to female genital mutilation. Those obstetric–gynaecological complications presenting at higher percentages in the sub-Saharan women of the present study sample were intrapartum and urgent caesareans, intense postpartum haemorrhage, tears (2nd and 3rd degree) with concurrent episiotomies (during the one same birth), severe risk of foetal distress, failed induction, and non-progressive labour. The medical records of the hospital where the study was performed did not include a record of the status of the genitalia. Therefore, it was not possible to make a comparison between mutilated and non-mutilated women. However, the bibliography consulted does point to obstetric problems related to FGM. In this sense, the World Health Organization, via a multi-country study, associates FGM with a high rate of complications both intrapartum and in the immediate postpartum timeframe. These complications exist for both mother and child, especially in cases involving infibulation [4]. This shows, objectively, that women with FGM are at greater risk of undergoing caesarean section, episiotomies, perineal discomfort, postpartum haemorrhage, and longer periods of admission than those not mutilated. Along similar lines, others authors [12,13,14] highlight that some of the most important sequelae of FGM for the health of such women are precisely parturition complications.

### 4.1. Caesarean, Instrumental Delivery, and Induced Labour

In the present study, caesareans were performed in 20.82% of the total number of participants, this being greater than the recommended standard set by WHO (<15%) [15]. Furthermore, according to the present data, caesareans were performed in the sub-Saharan women at a percentage twice that of the Spanish women. With regards to the type of caesarean, in the present study both intrapartum caesareans and urgent caesareans predominated considerably in sub-Saharan women, while for Spanish women it was the elective caesarean which predominated. Moreover, of the 18 cases of urgent caesareans reported in the present study, only one corresponded to the Spanish sample group, the remaining 17 were sub-Saharan women wherein the predominant reasons for performing an urgent caesarean were the severe risk of foetal distress (SRFD) and intense haemorrhage. These results coincide with those of Martínez García [16] who analysed the influence of maternal country of birth on healthcare assistance and pregnancy outcomes, showing a greater risk of urgent caesareans for sub-Saharan women. Additionally worthy of note is a systematic review into caesareans in immigrant residents of western industrialized countries, which reports that 69% of articles found differences between the latter group and the autochthonous women [17]. Those showing a higher risk were primiparous or multiparous African immigrant women, or from Southern Asia [17]. Along the same lines, a study conducted in Iceland in 2021 also established worse obstetric outcomes in migrant women from countries with a development index below 0.900 compared to women in the country [18]. In the bibliography addressing obstetric complications with female genital mutilation [4], it was shown that women with FGM types II and III were at a greater risk of undergoing caesarean than those not mutilated. Frega [19] and Indraccolo [20] also associated FGM with a greater risk of a prolonged expulsion stage and a greater incidence of caesareans. However, Davis [21] reports in his study that there was no difference in the caesarean section rate. These data point to the existence of differences in obstetric care among migrant women, but it is necessary to systematically record the state of the genitalia in order to be able to study comparative obstetrics outcomes in women who have undergone female genital mutilation.

In terms of instrumental births (forceps and vacuum), these were observed in 12.38% of all cases, which does comply with the standard of <15% [22]. However, labour induction was reported for 21.90% of total cases, more than double the standard set by the Spanish Ministry of Health at <10% [22]. The differences between the group of national women and the group of sub-Saharan women in terms of instrumental deliveries were not significant. This is in agreement with a study carried out in a population of Nigerian origin that migrated to Italy [23]. As with the present study, the Italian research confirms the existence of a greater number of tears in the migrant population. Sub-Saharan African women of Nigerian origin appeared more vulnerable and exposed to several adverse pregnancy outcomes [23]. Although no link can be made between the present results and FGM, it is necessary to point out that, in a systematic review by Berg and Underland [24], it was determined that women with FGM held a greater risk of prolonged parturition, induced labour and postpartum haemorrhage. 

### 4.2. Episiotomies and Tears 

The data from the present study reveal that episiotomies were performed in 25.03% of all cases; far removed from recommended levels (<15%) [22]. Nonetheless, the present study sample is found to present a lower percentage of episiotomies than those performed across Spain in 2005, which at that time ranged anywhere from 33% to 73%, according to data from Spanish Autonomous Communities [25]. Likewise, they also presented to a lesser degree than that revealed by a previous study carried out in the same hospital as the present one, which showed a presentation of episiotomies at around 50% of total births [26]. Furthermore, in a number of studies on immigrant women of sub-Saharan origin in a range of countries such as Norway [27] and Australia [28], such sub-Saharan women were seen to present episiotomies and tears (concurrently), to a greater degree. These data are in line with the Gudmundsdottir [18] study conducted in a migrant population from low-income countries, and with the study by Lorthe [29], who also found unjustifiable differences in obstetric outcomes in the native and migrant population. According to Wuest [30], intrapartum third degree tears affecting even the anal passage are common in women subjected to FGM. It is noteworthy that only seven of the 735 women from the present study had suffered third degree tears, all of whom were sub-Saharan. In the only documented case of FGM in the present study, the subject suffered a second degree paraurethral tear during parturition.

Finally, it should be noted that the present study shows significant differences in the type of delivery between native and sub-Saharan women. Specifically, late gestational losses could be related to poor follow-up of migrant women [31,32]. This poor follow-up is related to a greater difficulty in accessing health systems [31,32].

### 4.3. Limitations

The lack of genital status records in the electronic medical record is the major limitation of the study. The results of the present study do not show whether the obstetric complications observed in the participating sub-Saharan women are caused by FGM/A, non-optimal perinatal care, baseline conditions or those otherwise associated with gestation, sociocultural issues, or a combination of all or any of these factors. Nonetheless, despite such confusion bias, they do demonstrate this subset of the current Spanish population to be an obstetric high-risk group requiring special attention and care. Likewise, numerous clinical variables are seen to be positively or negatively associated with the prevalence of episiotomies, tears, and the remaining obstetric complications studied, yet they were not taken into account during the analysis of the present study data. These were variables such as: maternal posture during the expulsion stage, the use of epidural analgesia, the use of intrapartum oxytocin, gestational age, or foetal weight. The fact that upon cross-referencing certain variables no statistically significant differences were found, must be highlighted as a relevant result. These variables scarcely presented any data. Consequently, it is plausible to consider such findings as being due to, or influenced by, a lack in statistical data volume. Therefore, larger samples would be recommendable for future research in order to reduce such limitations. 

Although the types of bias detected in the present study may limit the interpretation of the data collected, as well as the capacity to extract solid conclusions regarding FGM/A, they do not interfere with the more than conclusive result that the sub-Saharan women studied presented a greater prevalence of obstetric complications compared to the Spanish women (regardless of whether FGM/A is considered one of the probable causes or otherwise). 

## 5. Conclusions

The present study revealed an under-recording of the assessment of the external genitalia prior to the delivery process. This resulted in no recording of the possible female genital mutilations present in the sample. The exposure of this under-recording led to an organizational change in the hospital to avoid the absence of this data in the medical records. Subsequent to the study, registry protocols were modified to ensure the inclusion of genitalia status in the electronic medical record. This represents an important advance for the generation of useful databases in the study of the influence of variables such as FGM on obstetric outcomes.

The comparative analysis of the data collected in the present study, on the differences in complications during parturition and the initial puerperium between sub-Saharan and Spanish women, provides us with an intimation on the sequelae that cultural differences and social determinants can impose on women’s health and integrity. The differences detected suggest differences in health care performance without clinical justification. It is therefore necessary to further study cultural barriers in order to address them. Many sub-Saharan women may become victims and/or survivors of FGM/A due to its high levels of prevalence, as registered in their countries of origin. Therefore, healthcare professionals must not only be aware of the issue, but also trained towards the acquisition of cultural competencies, so that they might be capable of acting in a systematic manner, via a multidisciplinary approach, focused fundamentally on preventing causative barriers. 

## Figures and Tables

**Figure 1 ijerph-19-10101-f001:**
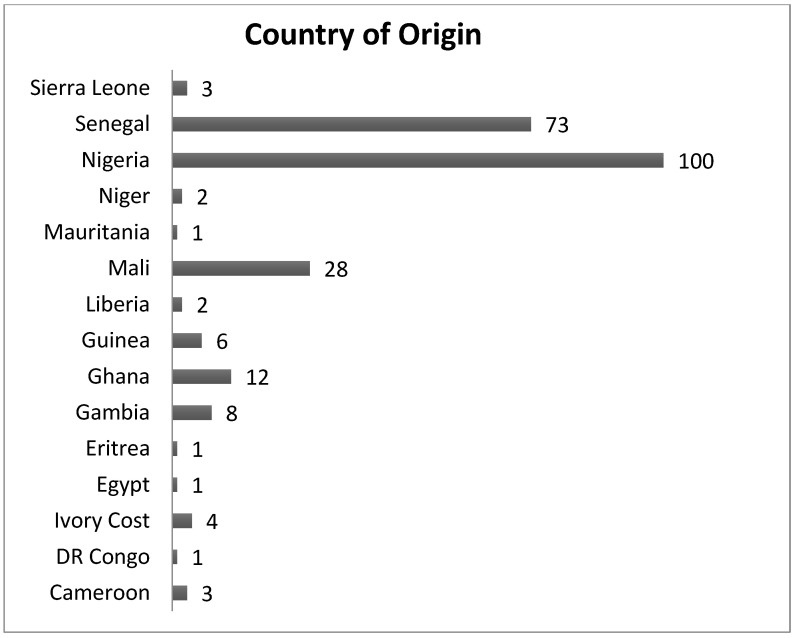
Number of study subjects according to country of origin.

**Table 1 ijerph-19-10101-t001:** The variable type of birth according to origin.

Type of Birth	Spanish n (%)	Sub-Saharan n (%)
Abortion	4 (0.82)	6 (2.45)
Elective caesarean	43 (8.78)	24 (9.80)
Intrapartum caesarean	33 (6.73)	35 (14.29)
Urgent caesarean	1 (0.20)	17 (6.94)
Eutocic birth	346 (70.61)	134 (54.69)
Type of birth unrecorded	0 (0)	1 (0.41)
Forceps	10 (2.04)	4 (1.63)
Vacuum	53 (10.82)	24 (9.80)
Total	490 (100)	245 (100)

**Table 2 ijerph-19-10101-t002:** Reasons for caesarean according to origin.

Reasons for Caesarean n = 153	Origin n (%)	
	Spanish	Sub-Saharan
**Reasons for elective caesarean n = 67**	43 (100)	24 (100)
Previous Caesarean (PC)	17 (39.5)	9 (37.5)
Intrauterine Growth Delay (IGD)	1 (2.3)	1 (4.2)
Prior Uterine Surgery	0 (0.0)	2 (8.3)
Unfavourable Obstetric Conditions	1 (2.3)	1 (4.2)
Multiple Gestation Pregnancy	3 (7.0)	0 (0.0)
Foetal Macrosomia + Maternal Infection	0 (0.0)	1 (4.2)
Non-reassuring Foetal Monitoring Status (NRFMS)	0 (0.0)	1 (4.2)
Maternal Condition (Severe Preeclampsia)	1 (2.3)	2 (8.3)
Occlusive Placenta Previa (OPP)	0 (0.0)	1 (4.2)
Breech Presentation	20 (46.5)	6 (25.0)
**Reasons for intrapartum caesarean n = 68**	33 (100)	35 (100)
Foetopelvic Disproportion (FPD)	3 (9.1)	1 (2.9)
Failed Induction (FI)	14 (42.4)	19 (54.3)
Non-Progressive Labor (NPL)	8 (24.2)	12 (34.3)
Moderate Risk of Foetal Distress (MRFD)	8 (24.2)	3 (8.6)
**Reasons for urgent caesarean n = 18**	1 (100)	17 (100)
Premature Abruption of Normally Implanted Placenta	0 (0.0)	1 (5.9)
Intense Haemorrhage	0 (0.0)	5 (29.4)
Umbilical Cord Prolapse	0 (0.0)	2 (11.8)
Uterine Rupture	0 (0.0)	1 (5.9)
Severe Risk of Foetal Distress (SRFD)	1 (100)	8 (47.1)

**Table 3 ijerph-19-10101-t003:** Reasons for inducing labour according to origin.

Reasons for Labor Induction N = 735	Origin n (%)	
	Spanish	Sub-Saharan
Intrauterine Growth Delay (IGD)	6 (1.22)	3 (1.22)
Gestational Diabetes (GD)	3 (0.61)	6 (2.45)
Hypertensive Disorders of Pregnancy (HDP)	4 (0.82)	1 (0.41)
Prolonged Pregnancy (PP)	18 (3.67)	7 (2.86)
Non-Reassuring Foetal Monitoring Status (NRFMS)	1 (0.20)	5 (2.04)
Non-Progressive Labor (NPL)	2 (0.41)	13 (5.31)
Oligohydramnios	7 (1.43)	2 (0.82)
Maternal Illness	0 (0.00)	2 (0.82)
Severe Preeclampsia	1 (0.20)	3 (1.22)
Risk of Foetal Distress (RFD)	2 (0.41)	3 (1.22)
Premature Membrane Rupture (PMR)	58 (11.84)	14 (5.71)
No Record of Induction (NRI)	388 (79.18)	186(75.92)
**Total**	490 (100)	245 (100)

**Table 4 ijerph-19-10101-t004:** Reasons for instrumental births according to origin.

Reasons for Instrumental Birth n = 91	Origin n (%)	
	Spanish	Sub-Saharan
Failed Induction due to GD	0 (0.0)	1 (3.6)
Failed Induction due to HDP	2 (3.2)	0 (0.0)
Failed Induction due to PP	5 (7.9)	0 (0.0)
Failed Induction due to Oligohydramnios	1 (1.6)	0 (0.0)
Failed Induction due to Severe Preeclampsia	1 (1.6)	2 (7.1)
Failed Induction due to PMR	11 (17.5)	4 (14.3)
NPL	43 (68.3)	15 (53.6)
NPL + RFD	0 (0.0)	2 (7.1)
NPL + RFD + Failed Induction due to NRFMS	0 (0.0)	1 (3.6)
RFD + Failed Induction due to NPL	0 (0.0)	3 (10.7)
**Total**	63 (100)	28 (100)

**Table 5 ijerph-19-10101-t005:** The variable episiotomy/tear/intact perineum according to origin.

Episiotomy/Tear/Intact Perineum N = 735	Origin n (%)	
	Spanish	Sub-Saharan
**Tear**	88 (17.96)	52 (21.22)
1st Degree Tear	57 (11.63)	28 (11.43)
2nd Degree Tear	23 (4.69)	10 (4.08)
3rd Degree Tear	0 (0.00)	6 (2.45)
“Cervical” Tear ^1^	1 (0.20)	2 (0.82)
“Multiple” Tears ^1^	0 (0.00)	1 (0.41)
“Vaginal” Tear ^1^	7 (1.43)	5 (2.04)
**Episiotomy**	107 (21.84)	47 (19.18)
**Episiotomy and Tear**	11 (2.24)	19 (7.76)
Episiotomy + 1st Degree Tear	3 (0.61)	10 (4.08)
Episiotomy + 2nd Degree Tear	5 (1.02)	3 (1.22)
Episiotomy + 3rd Degree Tear	0 (0.00)	1 (0.41)
Episiotomy + “Vaginal” Tear	3 (0.61)	5 (2.04)
**Intact Perineum**	284 (57.96)	126 (51.43)
**Not Recorded**	0 (0.00)	1 (0.41)
**Total**	490 (100)	245 (100)

^1^ No record was made in the patient notes of the exact degree of these three types of lesion, thus they were included separately in order to avoid selection bias.

## Data Availability

The data are available upon email request to the corresponding authors.

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
