# Peer review of "Obstetric Complications in Women from Sub-Saharan Africa—A Cross-Sectional Study"

_ijerph, 2022, doi:10.3390/ijerph191610101_

Round 1

Reviewer 1 Report

It is not enough to state because women originate from areas where FGM is done that for that reason that they have been mutilated

Important to report in the group study whether they had actually been mutilated and if so what grade of mutilation

Author Response

First, we would like to thank you for your comments on the submitted manuscript.

Secondly, we would like to respond to the comments made. The present study is a retrospective investigation. This research is influenced by the type of electronic records made by the hospital's obstetrics and gynecology professionals. In this sense, and up to the time of the study, the recording of the state of the genitalia in the computerized clinical history was not consistently performed. The lack of a record of the state of the genitalia makes it impossible to confirm the existence of mutilated women in the sample. This is originally reflected in the discussion and limitations of the study. It is also made explicit in the objective of the study. However, after comments and reading, we thought it necessary to state this situation more clearly in the section on results, discussion, limitations and conclusions. Efforts to clarify this aspect of the manuscript are shaded in yellow.

Reviewer 2 Report

The article entitled „Obstetric complications in women from sub-Saharan Africa. A cross-sectional study” raise up an important topic. Althought we live in XXI century, thus in some cultures are still preparing traditional practices which may have impact on human health. One of these is Female Genital Mutilation (FMG) in Sub-Saharan women and girls. The Authors of the study compared 245 women originated from the region where the FGM is practiced with 490 Spanish women in terms of obstetric complications.

The study have been conducted properly, the cited references are mostly new (due to 2022), however I have some remarks which should be clarified.

11.      Information on the age of FMG performance is missing.

22.      I have not noticed if there is information concerning: total elimination or harm female genitals?

Author Response

Dear reviewer, thank you very much for the effort to evaluate the present work. In response to the considerations made, it is necessary to inform that we do not have data corresponding to the existence or not of FGM in the selected sample. This is a retrospective study that relies directly on the electronic records of the patients' medical history up to the selected date. The study demonstrated under-recording of genitalia status in both women from countries where FGM is performed and in indigenous women. This under-recording substantially limits the scope for comparison of obstetric outcomes between women in both sample groups. However, the research team detected serious differences in obstetric outcomes according to place of origin. These limitations were reported in the discussion, limitations and conclusions section. However, in order to explain this aspect in more detail, we have chosen to include information in the results section. We have also clarified the information provided in the discussion, limitations and conclusions sections. Efforts to clarify this aspect of the manuscript are shaded in yellow.

Reviewer 3 Report

Dear author's

I was pleased to review your article "Obstetric complications in women from sub-Saharan Africa. A  crosssectional study". I have the following comments:

- What new information brings your study?

- Knowing this results how do you propose to improve outcomes in pregnant women originating from countries were they are at risk of FGM?

- It will be interesting to clarify the complication according FGM type.

- Minor English edits.

Author Response

First of all, we would like to thank you for your comments on this article. We will now respond to the comments made.

The present study reports the existence of serious inequalities in obstetric and gynecological care for women from sub-Saharan countries. The difference in obstetric outcomes may be due to multiple variables including FGM. However, the under-recording of genitalia status in both sub-Saharan and Spanish women prevented the detection of women with FGM in the sample. Despite not being able to clarify the specific cause of the differences, information is provided on the existence of these differences. The study makes explicit the existence of different obstetric outcomes depending on the place of origin. These inequalities have repercussions on women's health and their causes should be studied. This study provides evidence of their existence in a Spanish public hospital where, a priori, they should not exist. The reasons for these differences should be studied in subsequent studies. The study also provides evidence of the existence of an under-recording of the state of the genitalia of the women attended in the hospital. The study led to organizational changes and the obligation to record genitalia status in the hospital.

As for how to improve obstetric care, the present study shows a reality derived from unequal socio-health care. The first step to solve it is to detect it and this study mentions such diagnosis. The second step is to identify the causes of these inequalities. These causes may include the conditions in which these people are cared for, traditional practices that are harmful to women's health (such as female genital mutilation) or cultural differences. Demonstrating the existence of these inequalities is necessary to start working on them.

Finally, it is necessary to emphasize that the present study is a retrospective investigation. This research is influenced by the type of electronic records made by the hospital's obstetrics and gynecology professionals. In this sense, and up to the time of the study, the recording of the state of the genitalia in the computerized clinical history was not consistently performed. The lack of a record of the state of the genitalia makes it impossible to confirm the existence of mutilated women in the sample. This is originally reflected in the discussion and limitations of the study. It is also made explicit in the objective of the study. However, after comments and reading, we thought it necessary to state this situation more clearly in the section on results, discussion, limitations and conclusions. Efforts to clarify this aspect of the manuscript are shaded in yellow.

Round 2

Reviewer 3 Report

No comment’s for author’s.